# Phenotypes and rates of cancer-relevant symptoms and tests in the year before cancer diagnosis in UK Biobank and CPRD Gold

**Matthew Barclay** [1]*, **Cristina Renzi** [1,2], **Antonis Antoniou** [3], **Spiros Denaxas** [4], **Hannah Harrison** [3], **Samantha Ip** [3,5], **Nora Pashayan** [6], **Ana Torralbo** [4], **Juliet Usher-Smith** [3], **Angela Wood** [3,5,7,8,9,10], **Georgios Lyratzopoulos** [1]

1 Department of Behavioural Science and Health, Institute of Epidemiology and Healthcare, University College London, London, United Kingdom, 2 Faculty of Medicine, University Vita-Salute San Raffaele, Milan, Italy, 3 Department of Public Health and Primary Care, School of Clinical Medicine, University of Cambridge, Cambridge, United Kingdom, 4 Institute of Health Informatics, University College London, London, United Kingdom, 5 Victor Phillip Dahdaleh Heart and Lung Research Institute, University of Cambridge, Cambridge, United Kingdom, 6 Department of Applied Health Research, Institute of Epidemiology and Healthcare, University College London, London, United Kingdom, 7 British Heart Foundation Centre of Research Excellence, University of Cambridge, Cambridge, United Kingdom, 8 National Institute for Health and Care Research Blood and Transplant Research Unit in Donor Health and Behaviour, University of Cambridge, Cambridge, United Kingdom, 9 Health Data Research UK Cambridge, Wellcome Genome Campus and University of Cambridge, Cambridge, United Kingdom, 10 Cambridge Centre for Artificial Intelligence in Medicine, University of Cambridge, Cambridge, United Kingdom

* m.barclay@ucl.ac.uk

**Data Availability Statement:** Data sharing agreements and potential concerns around patient confidentiality prevent open sharing of the underlying data for this study. UK Biobank data can

## Abstract

Early diagnosis of cancer relies on accurate assessment of cancer risk in patients presenting with symptoms, when screening is not appropriate. But recorded symptoms in cancer patients pre-diagnosis may vary between different sources of electronic health records (EHRs), either genuinely or due to differential completeness of symptom recording. To assess possible differences, we analysed primary care EHRs in the year pre-diagnosis of cancer in UK Biobank and Clinical Practice Research Datalink (CPRD) populations linked to cancer registry data. We developed harmonised phenotypes in Read v2 and CTV3 coding systems for 21 symptoms and eight blood tests relevant to cancer diagnosis. Among 22,601 CPRD and 11,594 UK Biobank cancer patients, 54% and 36%, respectively, had at least one consultation for possible cancer symptoms recorded in the year before their diagnosis. Adjusted comparisons between datasets were made using multivariable Poisson models, comparing rates of symptoms/tests in CPRD against expected rates if cancer site-age-sex-deprivation associations were the same as in UK Biobank. UK Biobank cancer patients compared with those in CPRD had lower rates of consultation for possible cancer symptoms [RR: 0.61 (0.59–0.63)], and lower rates for any primary care consultation [RR: 0.86 (95%CI 0.85–0.87)]. Differences were larger for 'non-alarm' symptoms [RR: 0.54 (0.52–0.56)], and smaller for 'alarm' symptoms [RR: 0.80 (0.76–0.84)] and blood tests [RR: 0.93 (0.90–0.95)]. In the CPRD cohort, approximately representative of the UK population, half of cancer patients had recorded symptoms in the year before diagnosis. The frequency of non-specific presenting symptoms recorded in the year pre-diagnosis of cancer was substantially lower

be obtained from UK Biobank project site, subject to successful registration and application process. Further details can be found at https://www.ukbiobank.ac.uk/. CPRD Gold data can be obtained from CPRD, subject to protocol approval via CPRD's Research Data Governance Process. Further details can be found at https://cprd.com/data-access. All analysis code and phenotyping algorithms are available at https://github.com/MattEBarclay/ukb_cprd_cas_symptoms.

**Funding:** This work was supported by the International Alliance for Cancer Early Detection (ACED), a partnership between Cancer Research UK (C18081/A31373), Canary Center at Stanford University, the University of Cambridge, OHSU Knight Cancer Institute, University College London, and the University of Manchester. MB is supported by an ACED Pathway Award (EDDAPA-2022/100001). GL is supported by Cancer Research UK Advanced Clinician Scientist Fellowship C18081/A18180. Cristina Renzi acknowledges funding from Cancer Research UK — Early Detection and Diagnosis Committee (EDDCPJT/100018). AW is part of the BigData@Heart Consortium, funded by the Innovative Medicines Initiative-2 Joint Undertaking under grant agreement No 116074. AW is supported by the BHF-Turing Cardiovascular Data Science Award (BCDSA\100005). The funders had no role in study design, data collection and analysis, decision to publish, or preparation of the manuscript.

**Competing interests:** MB reports personal fees from Grail Inc for membership of an Independent Data Monitoring Committee, outside the submitted work. No other disclosures were reported.

among UK Biobank participants. The degree to which results based on highly selected biobank cohorts are generalisable needs to be examined in disease-specific contexts.

## Author summary

We develop symptom phenotypes and describe and compare rates of symptoms and blood tests before cancer diagnosis in patients drawn from a representative sample and from UK Biobank. We found that in the representative sample around half of patients had recorded symptoms in the year before cancer diagnosis, but that rates of recorded symptoms in UK Biobank participants were substantially lower. These differences primarily related to non-specific symptoms, but remained following adjustment for cancer site, age, sex, and socio-economic deprivation. The phenotypes for identifying symptoms we developed may be useful for other researchers working with UK primary care data. The differences in pre-diagnostic symptoms emphasise that the generalisability of results from cohort studies need to be examined in a disease-specific context, and support efforts for ensuring equitable participation in major cohort studies.

## Introduction

Early diagnosis is an important strategy for cancer control associated with improved clinical outcomes and patient experience [1–3]. Over 90% of patients in the UK and over 80% in the US are diagnosed after symptom onset [4–6], and understanding the risk of underlying cancer in symptomatic patients presenting to primary care is critical for achieving earlier diagnosis. Symptoms of possible cancer fall into two groups. First, so-called 'alarm' or 'red-flag' symptoms, for which national guidelines (for example in England and Scotland [7]) indicate that cancer should be suspected, and appropriate referral actions be considered. Overall, the positive predictive value of those symptoms exceeds 3% [8], which is deemed by the UK National Institute of Health and Care Excellence (NICE) to represent an appropriate risk threshold for fast-track investigation. Second, other symptoms with overall predictive value for cancer that is lower than 3%; around half of patients with cancer initially present with such non-specific or vague symptoms [9]. Yet positive predictive values are affected by the underlying risk, and values are generally substantially greater in older than younger individuals and men over women.

Early diagnosis strategies rely on primary care physicians being able to appropriately assess the risk of underlying cancer in symptomatic patients [10]. Several reviews and national guidelines summarise the evidence about the predictive value of common symptoms in either category and related positive predictive values [10–12]. Additionally, risk prediction models with a web interface exist that can support individualised risk prediction conditional on specific symptoms [13]. Currently available evidence chiefly pertains to sources of electronic health records data that do not include information on genetic risk [2,14–18]. The UK Biobank is one of the largest data sources where information from electronic health records and genetic risk are integrated. There is great interest in examining the contribution of genetic predisposition on cancer risk additional to other characteristics using UK Biobank data. Enabling such research requires development of appropriate phenotypes for symptoms of interest, and their distribution described and compared against other sources to guide interpretation. These considerations have motivated our study.

Many EHR datasets are representative of the reference population, including CPRD Gold [19]. Others cover a highly-selected cohort of patients, including current biobank cohorts such as UK Biobank. These selected cohorts are attractive for research due to the availability of linked genetic information and rich data on exposures during life-course in addition to EHR data on participants' healthcare use. However, the prevalence of health conditions can differ between EHR sources (e.g., [20,21]), either because of differing patient populations or because of differential completeness in the recording of these conditions in participating practices and EHR information systems used. UK Biobank participant health outcomes and demographic characteristics differ from the general population [22], and the impact these differences on the symptoms and tests in the year before a cancer diagnosis is unclear.

Given the above, this study had three aims. First, to develop harmonised symptom and blood test phenotypes for Read v2 and CTV3 coding systems. Second, to estimate the prevalence of 21 symptoms and eight blood tests in the year before cancer diagnosis in a representative cohort from CPRD Gold. Third, to compare rates of symptoms and blood tests in the year before diagnosis in cancer patients in CPRD and UK Biobank.

## Methods

The analysis had three main components: identifying comparable cohorts of cancer patients in UK Biobank and CPRD; identifying the same types of outcome events in each cohort; and calculating rates and adjusted comparisons between the cohorts.

### Data sources

CPRD Gold is an electronic health record (EHR) database of fully-coded primary care records from practices across the UK using the Vision software [19], with linkage to additional datasets including cancer registration data. Information on primary care consultations forms an inherent part of the EHR and is coded using Read v2, a clinical coding system primarily developed for primary care that has been used since 1985 [23]. The database contains data on more than 11 million patients from 1987 onwards [19], with an average follow-up of 5.1 years. Use of CPRD data in this study was approved by the UK Medicines and Healthcare products Regulatory Agency (MHRA) Independent Scientific Advisory Committee (ISAC Protocol number 18_299RMnA5), under Section 251 (NHS Social Care Act 2006).

UK Biobank is a prospective cohort study of around 500,000 people [24], recruited between the ages of 40 and 69 in 2006–2010. Around 200,000 UK Biobank participants have linked routinely collected primary care EHRs, recorded using either TPP (England, CTV3 coding, 71% of records), EMIS (Wales and Scotland, Read v2 coding, 20%), or Vision (England, Read v2 coding, 10%) systems. EHR data in UK Biobank are available from both before and after participants' recruitment to the study. Data from the UK Biobank Resource was provided under application number 64351; UK Biobank received ethical approval from the National Information Governance Board for Health and Social Care and the National Health Service North West Centre for Research Ethics Committee (Ref: 21/NW/0157). All participants provided informed consent at recruitment to the study for their data to be used for health-related research that was in the public interest.

**Study population.** We derived comparable cohorts within CPRD and UK Biobank for this analysis (Fig 1, S1 Fig). The CPRD cohort was drawn from a random sample of 1,000,000 patients in CPRD, while the UK Biobank cohort was drawn from those UK Biobank participants with available primary care records. Patients were included if they had a cancer diagnosis in 2006–2015 with at least a year of continuous (defined as no gaps in registration of more than 90 days) primary-care follow-up before diagnosis, and were aged 30–75 at diagnosis.

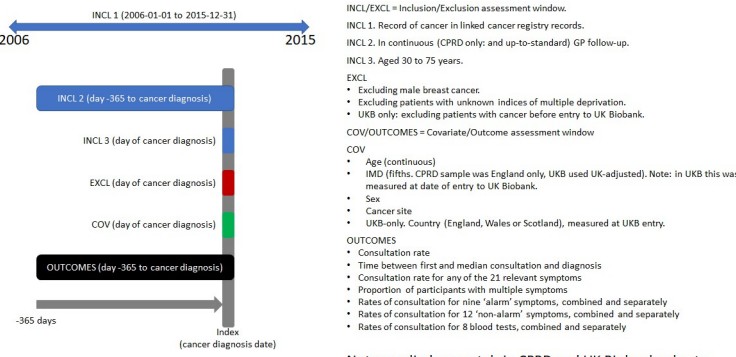

**Fig 1. Cohort definition chart for the UK Biobank and CPRD comparison cohorts.** A cohort inclusion/exclusion flowchart is given in S1 Fig. **CPRD comparison cohort**. CPRD Gold patients with a cancer diagnosed in 2006–2015, with at least a year of 'up-to-standard' primary care registration before diagnosis, and aged 30–75 at diagnosis to match UK Biobank cohort age ranges. **UK Biobank comparison cohort.** UK Biobank participants (with available electronic health records*) with a cancer diagnosed in 2006–2015 and at least a year of continuous GP record before diagnosis (including registrations with different practices if gaps in continuous registration were <91 days). *These patients belonged to the UK Biobank cohort sub-sample with available linked anonymous electronic health records data at the time of analysis.

Cancer diagnoses were sourced from the national cancer registry, and only the first cancer diagnosis post-2006 was considered. For UK Biobank patients, patients with cancer diagnoses before entry to the UK Biobank study were excluded.

**Cancer sites.** We considered all cancer sites combined (excluding non-melanoma skin cancer), and additionally the ten most common cancer sites in the UK Biobank cohort individually: breast, prostate, colorectal, lung (including mesothelioma), bladder, uterine, kidney, and upper GI cancers (stomach and oesophageal), and non-Hodgkin lymphoma (NHL) and melanoma, with all other sites considered as an 'other cancer sites' group. S1 Table gives ICD10 codelists for the sites included in each group.

## Outcomes

Outcomes were identified from coded electronic primary care record data using phenotypes developed for this study; free-text information was not available. We considered rates of primary care consultations, relevant symptoms, and blood tests (Table 1). These outcomes were identified from electronic health records using Read v2 and CTV3 phenotypes developed for this study (S1 Table).

One outcome related to the occurrence of primary care consultations (for any reason, without restriction to any specific recorded symptoms): the rate of primary care consultations in the year before cancer diagnosis.

Two outcomes related to primary care consultations for possible cancer symptoms (described subsequently) considered in aggregate (Table 1). First, the rate of primary care consultations for any of the relevant symptoms. Second, the proportion of patients with more than one relevant symptom in the year pre-diagnosis.

Ten outcomes related to 'alarm' symptoms for cancer (Table 1), selected based on previous evidence on their relatively high positive predictive value for cancer and clinical guideline recommendations for urgent specialist referral for suspected cancer [7], comprising the rate of consultation for any of nine 'alarm' symptoms overall, and for each alarm symptom individually.

**Table 1. Details of the different outcome measures considered.**

| Outcome | Measurement | Justification |
|---|---|---|
| Consultations for any reason. May include administrative and other non-clinical events. | Rate (per patient-year) | Capture general primary-care utilisation, expected to increase in year before diagnosis |
| Symptoms in primary care, as a composite | | |
| Consultations for any of 21 symptoms | Rate | Symptom consultations expected to increase more in year before diagnosis than general consultations |
| Proportion of patients with multiple distinct symptoms | Proportion | Hints partially at recording, i.e., if it is easy to record symptoms in one system then we might expect more patients with multiple symptoms |
| 'Alarm' symptoms in primary care | | |
| Consultations for any of 9 'alarm' symptoms | Rate | Symptoms that are known to be relatively strongly associated with a specific cancer. Typically identified in clinical guidance as indicating urgent assessment for suspected cancer in at least some age groups. |
| Abdominal lump | Rate | |
| Breast lump | Rate | |
| Change in bowel habit | Rate | |
| Dysphagia | Rate | |
| Haematuria | Rate | |
| Haemoptysis | Rate | |
| Jaundice | Rate | |
| Post-menopausal bleed | Rate | |
| Rectal bleeding | Rate | |
| 'Non-alarm' symptoms in primary care | | |
| Consultations for any of 12 'non-alarm' symptoms | Rate | Symptoms that are possibly related to a cancer diagnosis, perhaps of specific sites (e.g., abdominal pain and colorectal cancer) or of a range of sites (e.g., fatigue), but which are also common in primary care for other non-cancer reasons. |
| Abdominal bloating | Rate | |
| Abdominal pain | Rate | |
| Constipation | Rate | |
| Cough | Rate | |
| Diarrhoea | Rate | |
| Dyspepsia | Rate | |
| Dyspnoea (shortness of breath) | Rate | |
| Fatigue | Rate | |
| Night sweats | Rate | |
| Pelvic pain | Rate | |
| Nausea / vomiting | Rate | |
| Weight loss | Rate | |
| Blood tests in primary care | | |
| Any of the below blood tests | Rate | Especially for 'non-alarm' symptoms, blood tests in primary care may provide useful information for decision-making by GPs and indicate the possibility of cancer. |
| Haemoglobin concentration | Rate | |
| Platelet counts | Rate | |
| Haematocrit percentage | Rate | |
| Albumin | Rate | |
| Ferritin | Rate | |
| ESR | Rate | |
| CRP | Rate | |
| PV | Rate | |

Thirteen outcomes related to 'non-alarm' symptoms (Table 1), with weaker associations with possible underlying cancer and generally excluded from clinical guidelines on urgent referrals [7], comprising the rate of consultations for any of twelve 'non-alarm' symptoms overall and for each non-alarm symptom individually. These represent the non-alarm symptoms deemed of greatest importance [8,25,26].

Nine outcomes related to blood tests that can support the diagnostic process (Table 1), comprising the rate of any of eight blood tests overall, and the rate of each blood test individually.

**Outcome phenotypes.** Each of the 29 symptom and blood test outcomes required a harmonised phenotype appropriate for use across two different coding systems, CTV3 and Read v2. The Read v2 system includes ~88,000 codes, while CTV3 includes ~140,000. Primary care data in CPRD Gold is coded in Read v2, and there are existing Read v2 symptom phenotypes from previous studies which informed the development of phenotypes used in this study [8]. In contrast, the majority of UK Biobank primary care data is recorded in CTV3, and while there are prior CTV3 blood test phenotypes there were no symptom phenotypes [27].

Consequently, we developed harmonised Read v2 and CTV3 symptom phenotypes. Starting with an existing Read v2 phenotypes, we identified candidate CTV3 phenotypes by mapping from Read v2 to CTV3 using published mapping files available from NHS TRUD [28]. These identify codes for the same clinical concept in the two systems, but do not identify all relevant codes and occasionally identified concepts that were not closely related or were generic. Targeted term searches were applied to identify additional potentially-relevant CTV3 codes not identified by the mapping files. The proposed CTV3 phenotypes were reviewed by clinically qualified co-authors (CR and GL) to remove generic codes (e.g., the CTV3 code XM1XP for 'O/E–pain'), and codes not related to symptoms under investigation. Informal validation included examination of counts and rates of codes in different cuts of the dataset, and assessment of trends over time. The resulting phenotypes included 1,776 codes (1,227 CTV3, 549 Read v2, see S1 Table).

## Statistical analysis

We examined the demographic and cancer case-mix of each cohort, and described the number and proportion of participants with each outcome.

We applied Poisson models for all cancers combined (adjusted for cancer site, age, sex, and fifth of Indices of Multiple Deprivation (IMD)) and each cancer site individually (adjusted for age, sex, and IMD fifth) to examine differences in pre-diagnostic consultation rates and timing of consultations beyond those expected based on simple demographics, both overall and for individual symptoms. Age was parametrised using restricted cubic splines with knots at ages 50, 60 and 70.

The UK Biobank and CPRD cohorts were stored and analysed separately, and the reported results were calculated by an approach analogous to indirect standardisation. For each outcome, a Poisson model was initially fit in CPRD. The coefficients of this model were then applied in the UK Biobank cohort to calculate the outcome rate expected if the associations in UK Biobank were the same as CPRD. A Poisson model was then applied in the UK Biobank cohort to measure the ratio of the observed outcome rate against that expected based on CPRD. Robust standard errors were used to account for possible overdispersion.

The observed differences in rates of observed symptoms and blood tests prompted an exploratory analysis to examine whether differences were seen for all patients in UK Biobank or only for some geographical areas. We compared observed rates in UK Biobank participants

who lived in England, Wales, and Scotland at entry to UK Biobank (inferred from the assessment centre) against those expected based on CPRD.

**Multiple comparisons.** Model-based comparisons were only conducted where the UK Biobank and CPRD comparison cohorts each had at least ten participants with the outcome. Given 35 outcomes examined for all cancers and for eleven specific cancer sites, there were 420 potential comparisons of which 270 met minimum sample size criteria above. Analyses for UK Biobank participants' country added a further 105 comparisons. While we do not report p-values, we provide 95% confidence intervals and these intervals have not been adjusted for multiple testing. Given 525 comparisons we would expect around 25 comparisons where CIs do not cross 1 by chance, even if consultation rates were identical in each cohort. Care should be taken when interpreting specific comparisons, especially those based on relatively small numbers of consultations.

## Results

### Case-mix and demographics of the cohorts

UK Biobank (n = 11,594) and CPRD (n = 22,601) cancer patients in our analysis samples had overall similar age and sex distributions (Table 2), with higher levels of deprivation in CPRD. The cancer case-mix differed substantially. Prostate cancer in particular was overrepresented in UK Biobank (2,190, 19% of cancers) compared with CPRD (3,071, 14%). Breast cancer was also more common in UK Biobank (2,581, 22%) than CPRD (4,369, 19%), while lung cancer was less common (806, 7% vs 2184, 10%).

### Primary care consultations for any reason

Nearly all (>99.8%) of cancer patients in UK Biobank and CPRD had at least one primary care consultation in the year pre-diagnosis, although the mean number of consultations was higher in CPRD cancer patients (16.2) compared with UK Biobank cancer patients (14.3, Table 2). After adjusting for cancer site, age, sex, and deprivation, UK Biobank patients had 14% fewer consultations in the year before diagnosis (RR 0.86, 95% CI 0.85 to 0.87, Fig 2). Differences in overall consultation rates in the year before diagnosis were broadly consistent for each individual cancer site considered (Fig 3, all comparisons described in S3 Table).

### Symptomatic consultations and blood tests

Half of CPRD patients (54%, 12,187 of 22,601) had a recorded consultation for at least one of the 21 symptoms we considered in the year before diagnosis (Table 2); one in five (19%, 4,290) had a record of two or more of these symptoms. UK Biobank participants were considerably less likely to have symptom records, with just one third (36%, 4,164 of 11,594) having a record of any of the 21 examined symptoms and one in ten (9%, 1,059) reporting two or more symptoms. These differences persisted after adjustment for age, sex, IMD fifth and cancer site, with a rate ratio of 0.61 (95% CI 0.59 to 0.63) for symptomatic consultations in UK Biobank vs CPRD (Fig 2), and a rate ratio of 0.47 (95% CI 0.45 to 0.50) for the proportion of patients who have consulted for multiple different symptoms in the year pre-diagnosis.

Comparisons for individual symptoms and blood tests are summarised below. Additional results showing comparisons for each individual symptom and blood test and for each of the individual cancer site groups considered are given in S3 Table. Summary figures for each cancer site (showing all symptoms and blood tests) are given in S1 Text. Fig 3 provides an overview of all comparisons.

**Table 2. Population characteristics and distribution of symptoms, blood tests and primary care consultation patterns in CPRD and UK Biobank.**

| | UK Biobank cohort | | CPRD cohort | |
|---|---|---|---|---|
| **DEMOGRAPHICS** | **Count** | **(%)** | **Count** | **(%)** |
| Patients | 11,594 | | 22,601 | |
| Male | 5,683 | (49.0%) | 10,453 | (46.3%) |
| Deprivation | | | | |
| Least deprived | 3,594 | (31.0%) | 5,550 | (24.6%) |
| 2nd fifth | 2,557 | (22.1%) | 5,049 | (22.3%) |
| 3rd fifth | 2,088 | (18.0%) | 4,795 | (21.2%) |
| 4th fifth | 1,822 | (15.7%) | 3,898 | (17.2%) |
| Most deprived | 1,533 | (13.2%) | 3,309 | (14.6%) |
| | **Mean** | **(SD)** | **Mean** | **(SD)** |
| Age (years) | 64.1 | (6.9) | 59.4 | (11.2) |
| **NUMBER OF CANCERS BY SITE** | **Count** | **(%)** | **Count** | **(%)** |
| Any cancer | 11,594 | | 22,601 | |
| Breast cancer | 2,581 | (22.3%) | 4,379 | (19.4%) |
| Prostate cancer | 2,190 | (18.9%) | 3,071 | (13.6%) |
| Colorectal cancer | 1,238 | (10.7%) | 2,233 | (9.9%) |
| Lung cancer | 806 | (7.0%) | 2,184 | (9.7%) |
| Melanoma | 622 | (5.4%) | 1,011 | (4.5%) |
| NHL | 474 | (4.1%) | 787 | (3.5%) |
| Bladder cancer | 380 | (3.3%) | 709 | (3.1%) |
| Uterine cancer | 335 | (2.9%) | 630 | (2.8%) |
| Upper GI cancer | 306 | (2.6%) | 779 | (3.4%) |
| Kidney cancer | 272 | (2.3%) | 481 | (2.1%) |
| Other cancers | 2,390 | (20.6%) | 6,337 | (28.0%) |
| **CONSULTATIONS** | **Mean** | **(SD)** | **Mean** | **(SD)** |
| Consultations in the year before diagnosis | 14.3 | (9.7) | 16.2 | (11.5) |
| **POSSIBLE CANCER SYMPTOMS** | **Patients with 1 or more record** | **(%)** | **Patients with 1 or more record** | **(%)** |
| Any symptom | 4,164 | (35.9%) | 12,187 | (53.9%) |
| Multiple symptoms | 1,059 | (9.1%) | 4,290 | (19.0%) |
| 'Alarm' symptoms | 1,891 | (16.3%) | 5,115 | (22.6%) |
| Abdominal lump | 54 | (0.5%) | 185 | (0.8%) |
| Breast lump | 489 | (4.2%) | 1,888 | (8.4%) |
| Change in bowel habit | 111 | (1.0%) | 362 | (1.6%) |
| Dysphagia | 91 | (0.8%) | 372 | (1.6%) |
| Haematuria | 709 | (6.1%) | 1,054 | (4.7%) |
| Haemoptysis | 45 | (0.4%) | 178 | (0.8%) |
| Jaundice | 55 | (0.5%) | 174 | (0.8%) |
| Post-menopausal bleed | 160 | (1.4%) | 355 | (1.6%) |
| Rectal bleed | 242 | (2.1%) | 770 | (3.4%) |
| 'Non-alarm' symptoms | 2,789 | (24.1%) | 8,991 | (39.8%) |
| Abdominal bloating | 101 | (0.9%) | 346 | (1.5%) |
| Abdominal pain | 701 | (6.0%) | 2,642 | (11.7%) |
| Constipation | 208 | (1.8%) | 731 | (3.2%) |
| Cough | 900 | (7.8%) | 3,497 | (15.5%) |
| Diarrhoea | 216 | (1.9%) | 764 | (3.4%) |
| Dyspepsia | 445 | (3.8%) | 1,082 | (4.8%) |
| Dyspnoea | 636 | (5.5%) | 1,870 | (8.3%) |

*(Continued)*

**Table 2.** (Continued)

| | UK Biobank cohort | | CPRD cohort | |
|---|---|---|---|---|
| Fatigue | 247 | (2.1%) | 995 | (4.4%) |
| Night sweats | 14 | (0.1%) | 61 | (0.3%) |
| Pelvic pain | 9 | (0.1%) | 24 | (0.1%) |
| Nausea / vomiting | 100 | (0.9%) | 519 | (2.3%) |
| Weight loss | 48 | (0.4%) | 414 | (1.8%) |
| **BLOOD TESTS** | 7,469 | (64.4%) | 14,522 | (64.3%) |
| Full blood count | 6,316 | (54.5%) | 12,914 | (57.1%) |
| Haemoglobin concentration | 6,304 | (54.4%) | 12,903 | (57.1%) |
| Platelet counts | 6,294 | (54.3%) | 12,792 | (56.6%) |
| Haematocrit percentage | 6,232 | (53.8%) | 12,599 | (55.7%) |
| Acute phase response | 6,779 | (58.5%) | 12,940 | (57.3%) |
| Albumin | 6,578 | (56.7%) | 12,663 | (56.0%) |
| Ferritin | 1,658 | (14.3%) | 2,606 | (11.5%) |
| Inflammatory markers | 3,154 | (27.2%) | 7,037 | (31.1%) |
| ESR | 2,293 | (19.8%) | 4,895 | (21.7%) |
| CRP | 2,130 | (18.4%) | 4,707 | (20.8%) |
| PV | 247 | (2.1%) | 781 | (3.5%) |

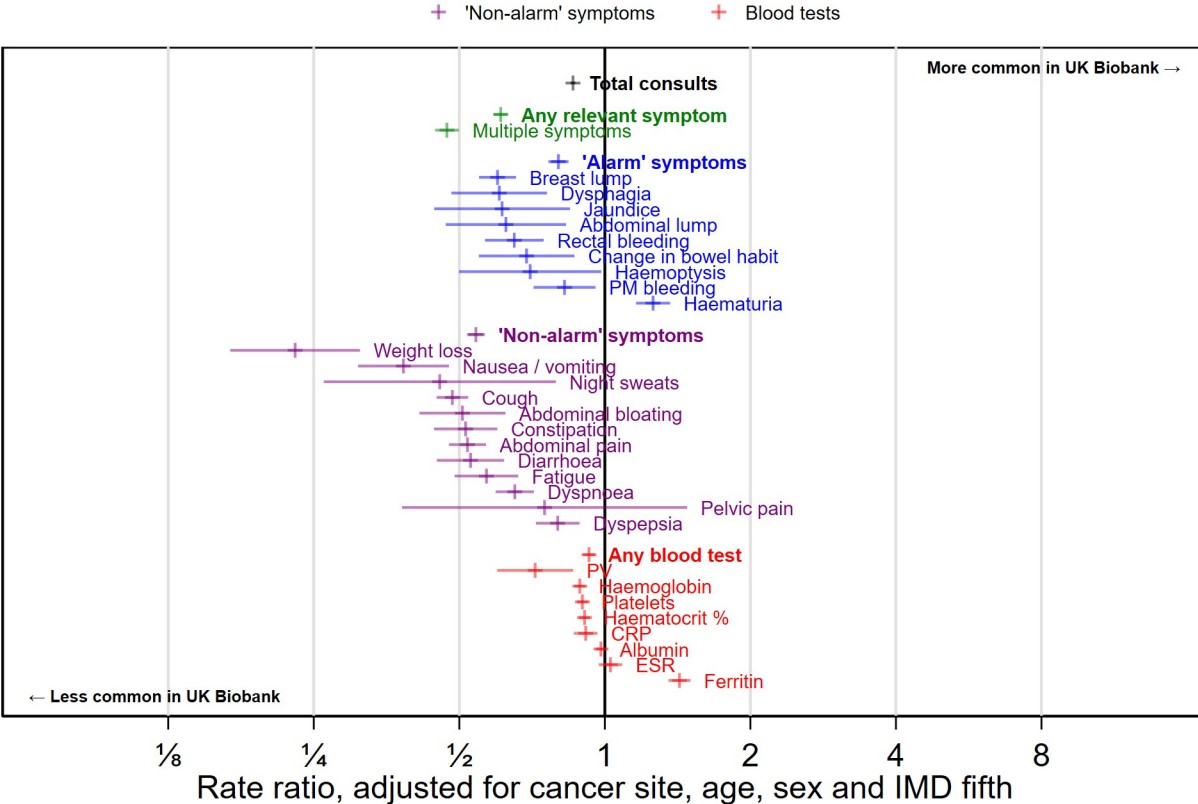

**Fig 2. Overall, all-cancers-combined, CPRD-UK Biobank comparisons of consultation, symptom and blood test rates, adjusted for age, sex, deprivation group (IMD fifth) and cancer site.**

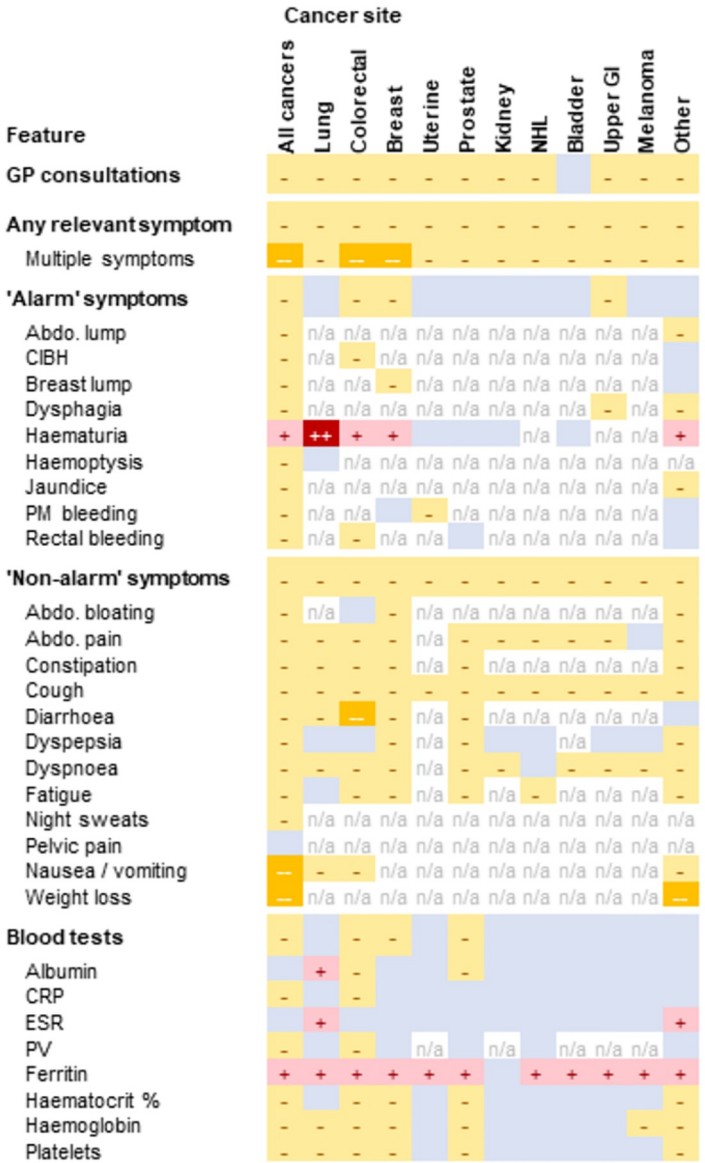

**Fig 3. Matrix of comparisons of consultations, symptoms, and blood tests between CPRD and UK Biobank in the year before a cancer diagnosis.** Dark yellow—means lower in UKB with upper bound below 0.5. Yellow—means lower in UKB with upper bound below 1. Red + means higher in UKB with lower bound above 1. Dark red ++ means higher in UKB with lower bound above 2. Blue without symbol means the CIs cross 1. n/a means comparisons were not performed due to small sample size.

**Records of consultations for 'alarm' symptoms.** 23% of CPRD patients and 16% of UK Biobank participants had at least one recorded consultation for an 'alarm' symptom recorded in the year before cancer diagnosis (Table 2), with corresponding adjusted (for age, sex, deprivation and cancer site) rate ratio of 0.80 (95% CI 0.76 to 0.84) for UK Biobank vs CPRD (Fig 2). This relatively small difference concealed heterogeneity in the direction of associations. Most symptoms were less common in UK Biobank patients although the size of the difference varied (8/9 alarm symptoms: abdominal lump; breast lump; change in bowel habit; dysphagia; haemoptysis; post-menopausal bleed; jaundice; and rectal bleeding). Haematuria was more

**Table 3. 'Alarm' symptom-cancer pairs in the UK Biobank and CPRD comparison cohorts.**

| Cancer site | Symptom | UK Biobank comparison cohort | | | CPRD comparison cohort | | | Adjusted rate ratio, UK Biobank vs CPRD | |
|---|---|---|---|---|---|---|---|---|---|
| | | Patients | | (% with symptom) | Patients | | (% with symptom) | | |
| | | Cancer | Symptom | | Cancer | Symptom | | RR | (95% CI) |
| Breast | Breast lump | 2,581 | 461 | (17.9%) | 4,379 | 1,771 | (40.4%) | 0.59 | (0.54, 0.65) |
| Colorectal | Rectal bleeding | 1,238 | 144 | (11.6%) | 2,233 | 487 | (21.8%) | 0.63 | (0.53, 0.76) |
| Bladder | Haematuria | 380 | 213 | (56.1%) | 709 | 455 | (64.2%) | 1.02 | (0.91, 1.14) |
| Uterine | Post-menopausal bleeding | 335 | 117 | (34.9%) | 630 | 270 | (42.9%) | 0.84 | (0.71, 1.00) |
| Kidney | Haematuria | 272 | 53 | (19.5%) | 481 | 103 | (21.4%) | 0.79 | (0.60, 1.05) |
| Upper GI | Dysphagia | 306 | 56 | (18.3%) | 779 | 241 | (30.9%) | 0.58 | (0.44, 0.76) |

common in UK Biobank than CPRD (RR 1.26, 1.15 to 1.36), with more extreme differences for cancer sites for which it is not a relevant symptom (e.g. lung RR 3.82, 95% CI 2.40 to 6.09, S3 Table) than those for which it is (e.g. bladder RR 1.02, 95% CI 0.91 to 1.14, Table 3).

Cancer-specific comparisons included six pairs of 'alarm' symptoms and associated cancers (Table 3). For three of these pairs (haematuria and both bladder and kidney cancers, and uterine cancer and post-menopausal bleeding and uterine cancer; and dysphagia and upper GI cancer (i.e. oesophageal and gastric cancer considered jointly), Table 3). For three of these comparisons (bladder cancer and haematuria, kidney cancer and haematuria, and uterine cancer and post-menopausal bleeding), after adjustment there was little evidence of a difference between UK Biobank and CPRD (Table 3). For the other three pairs (breast lump and breast cancer, rectal bleeding and colorectal cancer, and dysphagia and upper GI cancer), there were clear differences between the data sources with these symptoms being less common in UK Biobank patients than in CPRD.

**Consultations for 'non-alarm' symptoms.** 40% of CPRD and 24% of UK Biobank patients had at least one consultation for a 'non-alarm' symptom recorded in the year before cancer diagnosis (Table 2), with the corresponding adjusted (for age, sex, deprivation and cancer site) rate ratio being 0.54 (95% CI 0.52 to 0.56) for UK Biobank vs CPRD (Fig 2).

These patterns applied to all cancer sites (Fig 3), and most individual non-alarm symptoms considered (Figs 2 and 3). Yet, the scale of the differences varied. For example, the rate of consultations for abdominal pain was half as high in UK Biobank as in CPRD (RR 0.52, 95% CI 0.48 to 0.57, Fig 2), while for weight loss it was four times higher in CPRD (RR 0.23, 95% CI 0.17 to 0.31).

**Blood tests.** Sixty-four percent of both CPRD patients and UK Biobank participants had at least one blood test recorded in primary care in the year before cancer diagnosis. Across all cancers combined, and adjusted for age, sex, deprivation and cancer site, the rate ratio for blood tests was 0.93 (95% CI 0.90 to 0.95) in UK Biobank vs CPRD (Fig 2), and most individual blood tests had similar rate ratios to this overall comparison. Ferritin tests were the exception, with higher rates of testing in UK Biobank patients than in CPRD (RR 1.43, 95% CI 1.35 to 1.50).

## Differences in UK Biobank results by country

Exploratory analysis showed that UK Biobank participant differences from CPRD varied for the three UK nations (Fig 4, all country-specific comparisons given in S4 Table), with results for Scotland differing from those for England or Wales. There were larger differences in rates of recorded 'alarm' symptoms (RR 0.43, 95% CI 0.36 to 0.52 for Scotland UK Biobank

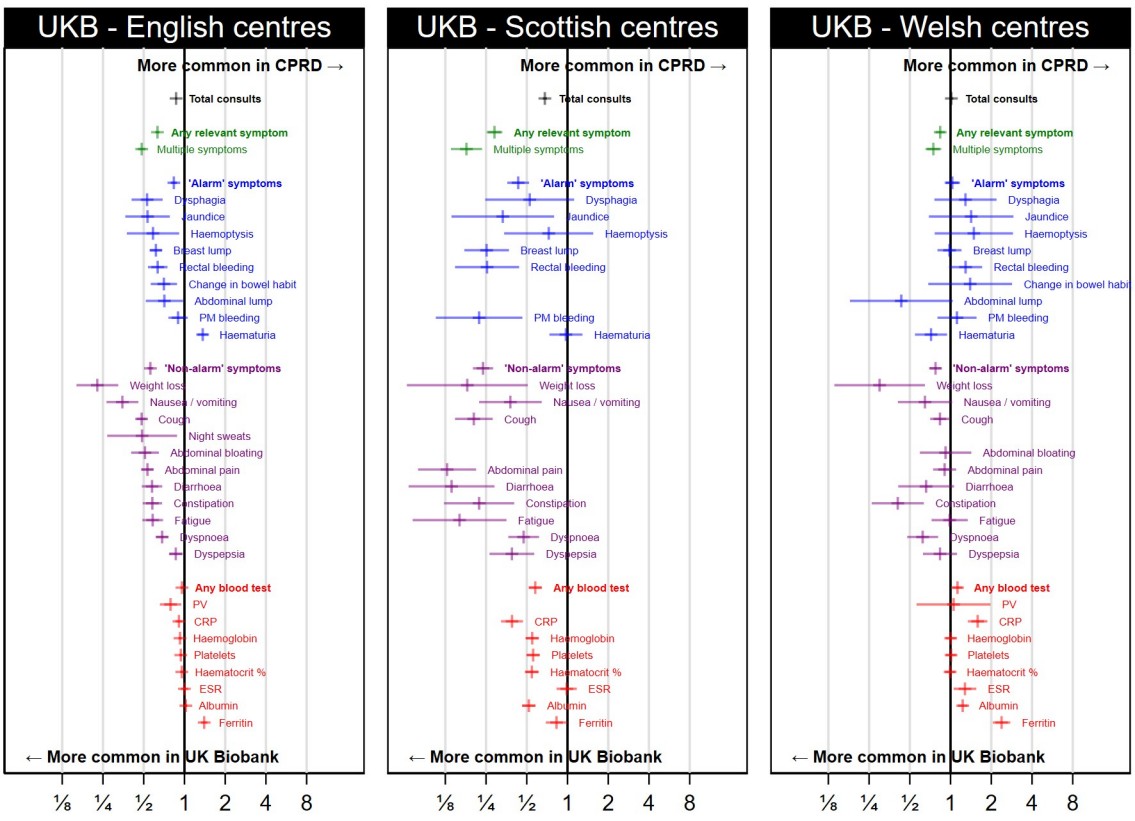

**Fig 4. Comparison of rates of symptoms and blood tests of UK Biobank participants in each nation against those in the CPRD cohort.** Some comparisons are suppressed due to small numbers.

participants vs CPRD, vs a corresponding RR 0.84, 95% CI 0.79 to 0.88 for England UK Biobank participants vs CPRD), and in rates of recorded blood tests (RR 0.58, 95% CI 0.52 to 0.65 for Scotland, vs RR 0.96, 95% CI 0.93 to 0.99 for England).

## Discussion

We created harmonised phenotypes for cancer-relevant symptoms and blood tests in two different primary care coding systems and compared their rates in the year before cancer diagnosis in participants of a cohort study and patients registered in UK general practice.

Nearly all cancer patients in either source had at least one primary care consultation in the year before their cancer diagnosis. Among CPRD cases, half had at least one consultation for one of the studied symptoms in the year before diagnosis, compared with one third in UK Biobank cases. As CPRD is approximately representative of the UK population, this likely means that a slight majority of incident cancer cases in the UK have recorded prodromal symptoms, but that a substantial fraction of patients either do not present to primary care with symptoms or do not have their symptoms recorded.

Differences in recorded alarm symptoms between the CPRD and UKB samples were generally small, with some variability between symptoms. A similar pattern of overall small differences was observed for use of common blood tests. However, differences in non-specific symptoms were far more pronounced.

## Comparisons with existing research

Pre-diagnostic prevalence of symptoms and blood tests in both UK Biobank and CPRD was of comparable magnitude to that reported by the English National Cancer Diagnostic Audit [29], although typically our study found lower prevalence of symptoms which may be due to symptoms being recorded in free-text information but not coded [30]. Other existing symptom studies found prevalence before diagnosis similar to our analysis of CPRD [31,32].

UK Biobank participants are not representative of the UK population [22], but estimates of associations of cardiovascular disease outcomes with known risk factors may be generalisable [20]. In contrast, the risk of adverse health outcomes in multimorbidity may be underestimated in UK Biobank [21]. In the light of existing literature, our results suggest the prevalence and predictive value of health states in selected cohorts should not be considered generalisable to broader populations; and that examination of representativeness needs to be considered in disease-specific contexts.

The findings highlight the risk of relying on single sources of EHRs when estimating prevalence or predictive values of less-specific symptoms (e.g., fatigue, abdominal pain) where it appears recording may be most variable. Despite calls for external validation [22], most studies of decision-support tools based on information on presenting symptoms have thus far been confined to single data sources, such as QResearch [9,10], CPRD [11], or THIN [12].

## Strengths and limitations

The key strengths of this study are: a) the use of high-quality longitudinal EHRs from sources with established previous use in research studies, b) the detailed process followed for the development of *de novo* harmonised phenotypes applicable across both sources of data, and c) the use of approaches akin to indirect standardisation to enable like-for-like comparisons of symptom occurrence in either cohort.

The main limitation is that, by necessity, the analysis could only examine symptoms recorded in the coded data. Free text information was not available for this study, and symptoms may not always be coded in the EHR. Calculating accurate estimates of symptom prevalence may require bespoke data collections such as the English National Cancer Diagnostic Audit [31]. Another challenge was the sample size of patients; cancer-site-specific and country-specific comparisons frequently needed to be suppressed due to the small number of patients with certain symptoms. One UK Biobank primary care data provider in England (Vision) did not provide information on patients who died before the data extract; this will have affected around 600 patients meaning there may be a slight underestimate of both cancer cases and symptoms in UK Biobank, but this could at most explain a small fraction of the differences between UKB and CPRD. Finally, there is possibly a small overlap of UK Biobank participants and CPRD patients, with some patients possibly appearing in both datasets; note however that this would necessarily reduce the size of any possible difference.

Further blood tests of relevance to cancer diagnosis exist that we did not examine, such as calcium and bilirubin. While there would be value in examining recording of further tests, we are reassured that most other blood tests had similar occurrence in both data sources.

Judging external validity of this study is hard due to the selection bias in UK Biobank. Differences between UK Biobank and CPRD will largely be driven by a mix of selection bias in UK Biobank and differences in recording of data due to the different general practice IT systems in use. When comparing other electronic health record datasets based on selected and representative samples, the mix of real differences in health due to selection and apparent differences due to completeness of recording will not be the same; dataset-specific studies will be required.

## Interpretation and implications

Observed differences in prevalence of pre-diagnostic features between UK Biobank and CPRD were principally concentrated in 'non-alarm' symptoms of relatively low positive predictive value, where risk of under-recording is greater [30]. However, this does not explain why possible under-recording of non-alarm symptoms should differ between the two sources.

Three hypotheses are worth exploring.

First, differences may truly reflect lower pre-diagnosis symptom prevalence in UK Biobank compared with CPRD cancer cases. This may be because of presentation earlier in the disease process (therefore with lesser symptom burden) or greater proportion of patients detected with asymptomatic disease (e.g., through screening for breast and bowel cancer, or through PSA testing). This may also explain the higher proportion of UK Biobank cancer patients with breast and prostate cancer. Ideally these hypotheses can be examined with information on diagnostic intervals, perhaps especially the time between symptom onset and first consultation with primary care, stage at diagnosis, and diagnostic route [33,34]. However, stage and route to diagnosis information is not yet available in UK Biobank linked cancer registry data.

Differences in comorbidity burden and lifestyle factors may also lead to real differences in rates of symptoms. UK Biobank participants have fewer comorbidities and lower rates of smoking than the general population [22]. Differences in smoking rates may be expected to lead to different rates of respiratory symptoms such as cough or dyspnoea, even among patients with smoking-related cancers. Among patients with colorectal cancer those with pre-existing comorbidities are less likely to present with rectal bleeding or change in bowel habit [35].

Second, there may be a systematic bias in the recording of symptoms in practices represented in UK Biobank and CPRD. Data on UK Biobank cases relates to practices using SystmOne/TPP, whereas data on CPRD cases to Vision. Computer system interface design is known to affect the way EHRs are used [36,37], possibly leading to differences in symptom coding, or how often symptoms are recorded as text rather than coded. For example, interface design could explain the observed excess haematuria codes for patients with lung cancer, which may be misrecorded haemoptysis if the interface suggests codes based on the first four letters. Such a systematic bias is a plausible explanation for the larger discrepancies observed when comparing CPRD vs UK Biobank participants from Scotland. This may also suggest caution when analysing electronic health records for UK Biobank participants from Scotland, as our results suggest they may be more likely to be missing information than records for England or Wales.

Third, our coding harmonisation may have been deficient in imperceptible ways, given differences in the coding systems. However, the exploratory analysis by country showed larger differences between CPRD cases and the Scotland component of UK Biobank cases (both coded Read v2), than between CPRD cases (coded in Read v2) and the England component of UK Biobank cases (chiefly coded in CTV3). This suggests differences are unlikely to be simply or solely due to problems with CTV3 phenotyping.

Overall, the findings offer reassurance but also a cause for concern. It is reassuring that within two UK populations of cancer patients, overall consultation patterns and recording of alarm symptoms and blood tests were broadly similar in the year before the diagnosis, indirectly cross-validating the usefulness of both sources for symptom-based research on cancer diagnosis. However, it is also concerning that on average there was a nearly two-fold difference in the frequency of recording of non-alarm symptoms, because predictive value estimates of such symptoms (for cancer) may be susceptible to bias, and because their accurate estimation is a research priority [38,39]. The potential direction of bias is hard to infer in advance of

empirical research, as it interacts with the degree of completeness in the recording of the same symptom among non-cases, but should be addressed by future research [40]. Given the variability in the encoding of symptoms, there is an urgent question about external validity of risk estimates (i.e., whether they can be transported from a given dataset to a population). The results highlight the need for further research of this kind across other sources of primary care electronic health records in the UK and health systems in other countries. Further efforts are needed to develop harmonised phenotypes that can be used across electronic health records datasets, as well as efforts to make use of free-text data from the electronic health record that may contain more complete information on symptoms. The findings support efforts for ensuring equitable participation in contemporary cohort studies collecting genetic, lifestyle and healthcare use data, such as the Our Future Health and All of Us cohorts.

## Supporting information

**S1 Fig. Cohort inclusion flowchart.**
(DOCX)

**S1 Table. Cancer groups and codelists in ICD10.**
(DOCX)

**S2 Table. Symptom and blood test codelists in Read v2 and CTV3.**
(DOCX)

**S3 Table. All feature comparisons between UK Biobank and CPRD.**
(DOCX)

**S4 Table. All feature comparisons between individual countries in UK Biobank and CPRD.** These comparisons relate to all cancer sites combined.
(DOCX)

**S1 Text. Plots of cancer-specific UK Biobank-CPRD comparisons.** Fig A. Breast cancer. Fig B. Prostate cancer. Fig C. Colorectal cancer. Fig D. Lung cancer. Fig E. Melanoma. Fig F. NHL. Fig G. Bladder cancer. Fig H. Uterine cancer. Fig I. Kidney cancer. Fig J. Upper GI cancer. Fig K. Other cancers.
(DOCX)

## Acknowledgments

Regarding the UK Biobank analysis, this research has been conducted using the UK Biobank Resource under application number 64351. Regarding the CPRD analysis, this study was approved by the UK Medicines and Healthcare products Regulatory Agency Independent Scientific Advisory Committee (ISAC Protocol number 18_299), under Section 251 (NHS Social Care Act 2006). This study is based in part on data from the Clinical Practice Research Datalink obtained under license from the UK Medicines and Healthcare products Regulatory Agency. The data is provided by patients and collected by the UK National Health Service (NHS) as part of their care and support. The interpretation and conclusions contained in this study are those of the authors alone.

## Author Contributions

**Conceptualization:** Matthew Barclay, Cristina Renzi, Antonis Antoniou, Nora Pashayan, Juliet Usher-Smith, Georgios Lyratzopoulos.

**Data curation:** Matthew Barclay, Cristina Renzi, Georgios Lyratzopoulos.

**Formal analysis:** Matthew Barclay.

**Funding acquisition:** Antonis Antoniou, Spiros Denaxas, Angela Wood, Georgios Lyratzopoulos.

**Investigation:** Matthew Barclay.

**Methodology:** Matthew Barclay, Spiros Denaxas, Hannah Harrison, Samantha Ip, Ana Torralbo, Angela Wood.

**Software:** Matthew Barclay.

**Supervision:** Matthew Barclay, Spiros Denaxas, Georgios Lyratzopoulos.

**Validation:** Matthew Barclay, Cristina Renzi, Georgios Lyratzopoulos.

**Visualization:** Matthew Barclay, Spiros Denaxas.

**Writing – original draft:** Matthew Barclay.

**Writing – review & editing:** Matthew Barclay, Cristina Renzi, Antonis Antoniou, Spiros Denaxas, Hannah Harrison, Samantha Ip, Nora Pashayan, Ana Torralbo, Juliet Usher-Smith, Angela Wood, Georgios Lyratzopoulos.

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
