## [Decision Letter · Decision Letter 0]

13 Jul 2023

PDIG-D-23-00086

Phenotypes and rates of cancer relevant symptoms and tests in the year before cancer diagnosis in UK Biobank and CPRD Gold

PLOS Digital Health

Dear Dr. Barclay,

Thank you for submitting your manuscript to PLOS Digital Health. After careful consideration, we feel that it has merit but does not fully meet PLOS Digital Health's publication criteria as it currently stands. Therefore, we invite you to submit a revised version of the manuscript that addresses the points raised during the review process.

Please submit your revised manuscript within 60 days Sep 11 2023 11:59PM. If you will need more time than this to complete your revisions, please reply to this message or contact the journal office at digitalhealth@plos.org. Please include the following items when submitting your revised manuscript:

We look forward to receiving your revised manuscript.

Kind regards,

Benjamin P. Geisler, M.D., M.P.H., F.A.C.P., M.R.C.P. (London), F.H.M.

Academic Editor

PLOS Digital Health

Journal Requirements:

1. Please provide separate figure files in .tif or .eps format only and remove any figures embedded in your manuscript file. Please also ensure that all files are under our size limit of 10MB.

Additional Editor Comments (if provided):

One of the reviewers argues that the alarm symptoms mentioned do not appear to be significant or well-validated. Please make sure that you address this comment to satisfy the reviewer's concern.

I would also ask that you explain the context of the study a bit better in the introduction: the positive predictive value of each single symptom from your reference seems to be quite low. What are relevant clinical and policy decisions (e.g., NHS strategies to address this or other clinical guidelines) regarding whether or not these signs and symptoms should be investigated (and how, e.g., stepwise or in a "shotgun" fashion)?

Finally, please add limitations to your study and not just discuss the differences between the databases but also the external validity and the clinical and policy implications.

Reviewers' comments:

Reviewer's Responses to Questions

**Comments to the Author**

1. Does this manuscript meet PLOS Digital Health’s publication criteria? Is the manuscript technically sound, and do the data support the conclusions? The manuscript must describe methodologically and ethically rigorous research with conclusions that are appropriately drawn based on the data presented.

Reviewer #1: Yes

Reviewer #2: Yes

Reviewer #3: No

2. Has the statistical analysis been performed appropriately and rigorously?

Reviewer #1: Yes

Reviewer #2: Yes

Reviewer #3: No

3. Have the authors made all data underlying the findings in their manuscript fully available (please refer to the Data Availability Statement at the start of the manuscript PDF file)?

Reviewer #1: Yes

Reviewer #2: Yes

Reviewer #3: Yes

4. Is the manuscript presented in an intelligible fashion and written in standard English?

Reviewer #1: Yes

Reviewer #2: Yes

Reviewer #3: Yes

5. Review Comments to the Author

Reviewer #1: 1. I wonder what was the reasoning behind using both hemoglobin concentration and hematocrit, as they are both indices of anemia? Were other common labs such as serum sodium (SIADH), calcium (NHL, multiple myeloma, lung cancer - hypercalcemia of malignancy), blood urea nitrogen (sensitive marker of GI bleeding), bilirubin (liver metastases, pancreatic cancer) available in both databases for comparison? 

2. Table 1: Night sweats and especially unintentional weight loss should generally be considered alarm symptoms. 

3. Table 2: Authors can consider presenting this data as a histogram or a bar chart (side-by-side comparison bars between CPRD and UKB) for better visibility. 

4. Fig 3: Authors may consider moving fig 3 to supplement as it does not add to the readability of the data presented. 

5. Fig 4: any possible explanation why Welsh centers were different from that of Scottish and English, i.e. Welsh centers were more similar to CPRD (general UK population)? Is the Welsh population not well-represented in the UKB?

6. [line 266] Sentence should not start with a number: 64%. Instead, it should state: Sixty-four percent. 

7. [line 313-314] Please clarify “non-specific exposures” in this context. 

8. Although it was not one of the primary aims of this study, it would be interesting to look for associations between nonspecific symptoms and malignancies for which there is no established screening (other than breast, CRC and prostate) and then compare them in aggregate between the two databases.

9. If numbers allow, would it be possible to further explore different combinations of symptoms and labs and see if there was a specific combination of these covariates that is associated with certain cancer diagnoses in the following year? For example, anemia and bloating together in prediction of GI malignancy as opposed to each of these covariates separately. 

In summary, this is a comprehensive study that tried to address an important problem of healthcare data extraction by harmonising EHRs that utilize different coding systems. The authors recognize that coded data offer limited insight and can lead to likely erroneous associations (hematuria and lung cancer). Analyses of unstructured data (free text from clinical notes), where available, by using natural language processing algorithms in the future will most likely be able to uncover a wealth of additional information and this should also be one of the concluding remarks and suggested future directions of this study.

Reviewer #2: Thank you for the opportunity to review this manuscript. Overall I thought the study was very well conducted and clearly presented. 

The study clearly shows differences in the recording of symptoms prior to cancer diagnosis between CPRD and UK Biobank. The methods are clearly explained. There are several potential causes for this (other than genuine differences in presentation) that occurred to me when reading the results (UKB participants more likely to be diagnosed through screening, differences in user-interface between systems making coding of symptoms more/less likely, etc.), but each of these were well covered in the discussion (which I thought was excellent and dealt with them in a very balanced way). 

I have some minor comments/questions. I realise some of these may/may not be possible with the available data (and don't consider them critical to the integrity of the study). 

1. I note that for the UK Biobank population the authors retained those participants from practices using the Vision system. My understanding is that there is (or was?) a data quality issue with these participants (missing data for those participants who had died - bottom of page 12 here https://biobank.ndph.ox.ac.uk/showcase/showcase/docs/primary_care_data.pdf). Can the authors comment on this, and justify their inclusion in the dataset if this is the approach taken. 

2. Is it possible to identify 'suspected cancer' referrals from either dataset? If so, a comparison of these might help to understand if differences in coding (rather than patients reporting) symptoms is responsible for the differences. (i.e. if suspected cancer referrals were similar, but symptoms lower in UKB, this might suggest that the issue is with lower coding rates within certain systems). As the authors imply, it is a shame that 'route to diagnosis' and stage information is not yet available. 

3. Can the authors confirm their definition of a 'consultation'. Is this any date on which 'event' codes were entered (and therefore are likely to be consultations but may be other contacts/admin events/coding of discharges etc.)?

Overall I thought this was an excellent description of an interesting study with a measured interpretation which was very mindful of the potential pitfalls in the data and the multiple possible explanations for the differences observed.

Reviewer #3: The “alarm” symptoms mentioned do not appear to be significant or well-validated, and even such a validation would likely be specific to a given database. It is also unclear how often these non-specific symptoms are present in an age-specific cohort manner. To then take this non-specific set of findings and compare them to another database with non-specific findings seems dubious at best. 

Comparing vague symptoms between two databases seems fraught. It is likely not particularly meaningful, except to make the case that these databases should not be combined for a joint analysis of this type. That could be the only outcome of this analysis, and I am not certain why that should be published in this journal or any journal.

6. PLOS authors have the option to publish the peer review history of their article (what does this mean?). If published, this will include your full peer review and any attached files.

**Do you want your identity to be public for this peer review?** For information about this choice, including consent withdrawal, please see our Privacy Policy.

Reviewer #1: No

Reviewer #2: No

Reviewer #3: No

---

## [Decision Letter · Decision Letter 1]

5 Oct 2023

Phenotypes and rates of cancer relevant symptoms and tests in the year before cancer diagnosis in UK Biobank and CPRD Gold

PDIG-D-23-00086R1

Dear Dr Barclay,

We are pleased to inform you that your manuscript 'Phenotypes and rates of cancer relevant symptoms and tests in the year before cancer diagnosis in UK Biobank and CPRD Gold' has been provisionally accepted for publication in PLOS Digital Health.

Best regards,

Benjamin P. Geisler, M.D., M.P.H., F.A.C.P., M.R.C.P. (London), F.H.M.

Academic Editor

PLOS Digital Health

Reviewer Comments (if any, and for reference):

Reviewer's Responses to Questions

**Comments to the Author**

1. If the authors have adequately addressed your comments raised in a previous round of review and you feel that this manuscript is now acceptable for publication, you may indicate that here to bypass the “Comments to the Author” section, enter your conflict of interest statement in the “Confidential to Editor” section, and submit your "Accept" recommendation.

Reviewer #1: All comments have been addressed

Reviewer #2: All comments have been addressed

2. Does this manuscript meet PLOS Digital Health’s publication criteria? Is the manuscript technically sound, and do the data support the conclusions? The manuscript must describe methodologically and ethically rigorous research with conclusions that are appropriately drawn based on the data presented.

Reviewer #1: Yes

Reviewer #2: Yes

3. Has the statistical analysis been performed appropriately and rigorously?

Reviewer #1: Yes

Reviewer #2: Yes

4. Have the authors made all data underlying the findings in their manuscript fully available (please refer to the Data Availability Statement at the start of the manuscript PDF file)?

Reviewer #1: Yes

Reviewer #2: Yes

5. Is the manuscript presented in an intelligible fashion and written in standard English?

Reviewer #1: Yes

Reviewer #2: Yes

6. Review Comments to the Author

Reviewer #1: The authors have comprehensively addressed all of my comments and concerns.

Reviewer #2: All comments addressed. No further comments/suggestions.

7. PLOS authors have the option to publish the peer review history of their article (what does this mean?). If published, this will include your full peer review and any attached files.

**Do you want your identity to be public for this peer review?** For information about this choice, including consent withdrawal, please see our Privacy Policy.

Reviewer #1: No

Reviewer #2: No
